# Temporal Change in Iron Content of Vegetables and Legumes in Australia: A Scoping Review

**DOI:** 10.3390/foods11010056

**Published:** 2021-12-27

**Authors:** Erica Eberl, Alice Shimin Li, Zi Yin Joanne Zheng, Judy Cunningham, Anna Rangan

**Affiliations:** 1Charles Perkins Centre, The University of Sydney, Camperdown, NSW 2006, Australia; eebe9552@uni.sydney.edu.au (E.E.); shli8956@uni.sydney.edu.au (A.S.L.); zzhe5033@uni.sydney.edu.au (Z.Y.J.Z.); 2Curtin School of Population Health, Curtin University, Kent Street, Bentley, WA 6102, Australia; judy.cunningham@curtin.edu.au

**Keywords:** iron content, vegetable, legume, food composition, Australia

## Abstract

Studies in UK and US have reported a temporal decline in the iron content of plant-based foods. Limited research on this topic has been conducted in Australia. The aim of this scoping review was to provide a comprehensive evaluation on the temporal change in iron content of Australian vegetables and legumes from 1900 onward. A systematic search of electronic databases, websites, backward reference searching, and Australian food composition tables was conducted. A total of 34 articles and six versions of Australian food composition databases published between 1930s to 2021, were included in this review. Overall, iron content of vegetables and legumes were assessed at limited time points and geographical origin, cultivars, sampling and analytical techniques varied across studies. The majority of vegetables had similar iron content between two or more timepoints but decreases of 30–50% were noted for sweet corn, red-skinned potatoes, cauliflower and green beans while increases of 150–300% were seen for Hass avocadoes, mushrooms and silverbeet. More pronounced reductions in iron content were observed for legumes, with higher and more variable values reported pre-2000 compared to recent years. Due to limited data and variations in sampling and analytical techniques, no definitive conclusions could be established. As plant-based diets are becoming more popular, consistent monitoring of the nutrient composition of staple plant-based foods is strongly recommended.

## 1. Introduction

Iron is commonly considered a mineral of concern for deficiency in people following vegetarian and vegan diets. While eliminating meat can be accomplished with minimal effect on total iron intake [1,2,3], vegetarian diets are associated with an increased consumption of phytate-containing vegetables, legumes and grains which hinder iron bioavailability [3,4]. A dependence on the less efficiently absorbed non-haem form as well as the presence of dietary components that inhibit intestinal solubility and absorption of iron [5,6] increases the likelihood of low iron stores in people consuming plant-based diets [5]. Although no recent data is available on the prevalence of iron deficiency in Australia for those following a plant-based diet, the prevalence of iron deficiency in adult women, between 25–50 years of age, is estimated to be 20% [7].

There is a growing interest towards plant-based diets for health, ethical, environmental, and religious reasons [8]. Recent market research has shown a rise in the practice of vegetarianism in Australia, with the proportions of population consuming a plant-based diet increasing from 9.7% in 2012 to 12.1% in 2018 [9]. Based on the most recent national dietary survey, approximately one in eight Australians have inadequate iron intakes, with the highest at-risk group being females aged 14 to 50 years at 40% [10]. As females are twice as likely to be vegetarian or vegan than males and have higher iron requirements [11], it is particularly important to consider the sources of iron intake when changing to a plant-based diet.

There is concern that the mineral content of plant-based foods has declined in the past few decades. A 1997 study comparing nutrient data of vegetables and fruits in two UK food composition tables analysed in 1930s and 1980s found a significant decline in the levels of magnesium, calcium, copper, sodium, potassium and iron in those foods over time [12]. Since then, studies comparing historical US food composition data have also noted a marked temporal decrease in mineral content of plant-based foods [13,14]. A potential explanation for the apparent decrease in nutrient levels of horticultural produce has been proposed, linking to the idea of a “dilution effect” whereby cultivar selection for increased crop yield results in an accumulation of carbohydrate (starch, sugar, fibre) without a proportional increase in mineral content [14].

There has been limited research examining the trend in mineral content of plant-based foods in Australia. An investigation by Food Standards Australia New Zealand (FSANZ) observed no significant temporal change in six studied minerals, including iron, for fruits and vegetables sampled in the 1980s and early 2000s [15]. No recent studies have comprehensively assessed the temporal changes in iron content of Australian plant-based foods, and thus the research question was to examine whether the iron content of vegetables and legumes available in Australia has changed over time.

Using a scoping review approach, the aims of this paper were to firstly, examine the temporal changes of iron content in vegetables and legumes drawing on scientific and grey literature, and secondly, to explore any iron content changes for these foods in the national food composition databases over time. Findings from this study are particularly relevant in the current environment with growing numbers of people adopting a plant-based diet.

## 2. Materials and Methods

This review is based on the Arksey and O’Malley [16] framework and aligns with the Joanna Briggs Institute Scoping Review approach [17] and the Preferred Reporting Items for Systematic Reviews (PRISMA) extension for Scoping Review checklist [18].

### 2.1. Eligibility Criteria

To be included in this review, articles were required to report numerical data on the iron content of Australian legumes or vegetables focusing on the varieties from the 5 highest iron-contributing vegetable groups (potatoes; fruiting vegetables (excluding tomato); leaf and stalk vegetables; brassica vegetables; and peas and beans) identified in the Australian Health Survey 2011-12 (Appendix A). This included iron content from edible portions of crops, in a raw or dried form. As a further requirement for legume data, reported content should be from the edible fruit, seed or grain of legume including the embryo or de-hulled components, provided more than half of the legume fruit was analysed. All literature written in English and published between 1900 to 2020 were considered.

Papers were excluded if they did not report iron nutritive values, analysed components of vegetables or legume plant that were not commonly used for human consumption or were not sourced in Australia, only studied legume- or vegetable-based products (such as flour, pasta, plant-based milk, tofu), or dishes that involved more than one single type of vegetable or legume (mixed salad). Studies investigating the effect of a treatment variable that would unnaturally alter the iron content, for example heavy metal pollution, without including a control group were also excluded.

### 2.2. Information Sources and Search

To identify potentially relevant articles, the following literature databases were searched systematically from inception dates (year 1900 as earliest) to 10 December 2021: MEDLINE (Ovid), EMBASE (Ovid), Global Health (Ovid), Scopus, Web of Science CAB Abstracts and Core Collections, ANR Index and Archive (Informit). The search strategies were revised with the assistance of an academic liaison librarian from the University of Sydney and further refined through team discussion. The final search strategies for MEDLINE searches can be found in Appendix A. Final search results were exported into EndNote X9 (Clarivate Analytics), and duplicates were removed.

A grey literature search was performed through hand-searching of websites of relevant organisations: Australian Export Grains Innovation Centre, Bean Growers Australia Limited, Department of Primary Industries and Regional Development’s Agriculture and Food Division, FSANZ, Grains and Legumes Nutrition Council (GLNC), Grain Industry Association of Western Australia, Grains Research and Development Corporation, Horticulture Innovation Australia Ltd., Pulse Australia. FAO/INFOODS Global Food Composition Database for Pulses-version 1.0 was also screened for relevant data. Up to the first 100 results of keyword string searches created by the combination of keywords representing three main topics of interest (e.g., potato AND iron content AND Australia) were screened on Google and Trove (National Library of Australia database) in November 2020. Backward reference searching of included records was conducted.

Six versions of reference food composition databases including the Composition of Foods Australia (CoFA) 1989, Nutrient Tables for Use in Australia (NUTTAB) 1991-2, NUTTAB 1995, NUTTAB 2006, NUTTAB 2010 and Australian Food Composition Database-Release 1.0 (2019) were examined to gather publicly accessible data on the iron contents of vegetables and legumes. Reference food composition databases published prior to 1989 were not included as these consisted largely of values obtained outside of Australia.

### 2.3. Selection of Sources of Evidence

All articles were independently screened by two reviewers [ASL and EE], first by title and abstract and then by full text. Disagreements in the literature selection process were resolved by consensus or through discussion with a third reviewer [AR].

### 2.4. Data Charting Process and Data Items

The data charting form was developed for this review and two reviewers [ASL and EE] charted the data on vegetables and legumes independently. Results were discussed and the data-charting forms were continuously updated in an iterative process to summarise key characteristics of included studies and relevant findings. The following data were charted: authorship, publication year, type of vegetable/legume, iron content (all converted to mg/100 g), sampling details (date and location, sample number, study method), and analytical method. For papers that did not specify the year of growth or sampling, the date was taken as the year of publication. Efforts were made to contact the author or affiliated organisation regarding any missing data in the included studies.

### 2.5. Synthesis of Results

The charted data were summarised within the individual vegetable and legume groupings to examine changes in iron content over time if multiple timepoints were available. For data obtained from the six Australian food composition databases, changes in iron content by vegetable or legume variety were examined between databases published between 1989 to 2019.

## 3. Results

### 3.1. Study Selection

A total of 19,538 articles were identified from literature databases, food composition tables, and grey literature searches (Figure 1). After removal of duplicates, 13,198 articles were screened based on title and abstract, which excluded 12,933 articles. Full text screening of 265 articles excluded a further 225 articles, mostly due to not analysing Australian vegetables or legumes, not reporting iron content with units of measure, or analysing a non-edible component of the vegetable or legume. In this case, 18 studies were excluded because they were unable to be retrieved after contacting library staff, affiliated organisations, and journals. The remaining 40 articles were considered eligible; 34 were primary analyses and 6 were reference food composition databases.

### 3.2. Study Characteristics

In this case, 34 primary analysis reports sampled Australian grown or sourced vegetables and legumes between 1930 and 2021, with the majority of research conducted between 1980 and 2010 (Table 1). These studies provided iron values for a total of 56 vegetable types/cultivars within the top 5 iron-contributing vegetable groups and 21 legume types/cultivars. Of these, 16 vegetable and 8 legume types were analysed at two or more time points.

Sampling varied among the studies, with most using a single or composite sample and reporting no statistical variability. The growing regions for vegetables and legumes were all within Australia but exact locations were not reported for all studies and nor were factors such as soil and fertiliser type. The majority of samples were collected in field, or at retail outlets although a small number were grown in greenhouse or in experimental conditions using an open-top or growth chamber. Analytical techniques used in the included studies differed over the years from thiocyanate colorimetry in the 1930s, atomic absorption spectroscopy (AAS) between 1970–1990, and inductively coupled plasma-mass spectrometry (ICP-MS) or inductively coupled plasma atomic emission spectroscopy (ICP-AES) from 2000 onwards. Data are summarised in Table 1.

### 3.3. Changes in Iron Content Based on Scoping Review Data

Temporal trends in iron content of vegetables and legumes, assessed using data from at least two time points (Table 1), are summarised in the sections below.

#### 3.3.1. Potatoes

Pale-skinned and red-skinned potatoes were considered separately, as was carried out in food composition databases. For pale-skinned potatoes, iron content remained relatively stable between three timepoints; 1982 (Sebago 0.6 mg/100 g) [26], 2000 (Coliban 0.5 mg/100 g) [15] and 2007–2008 (Shepody 0.7 mg/100 g) [36]. For red-skinned potatoes, a decrease was observed from 0.5 mg/100 g in 1982 (Pontiac) [26] to 0.2 mg/100 g in 2000 (Desiree) [15] although the cultivars differed between the timepoints.

#### 3.3.2. Stem and Leafy Vegetables

An apparent increase in iron content of silverbeet was observed across three time points from 1.9 mg/100 g in 1938 [19] to 2.3 mg/100 g in 1983 [30] and 3.0 mg/100 g in 2007–2008 [36]. Iron content of lettuce was reported over three time points and ranged from 1.0 mg/100 g in 1938 (unspecified variety) [19], to 0.6 mg/100 g in both 1983 [30] and 2000 (common/iceberg) [15]. For celery, iron content remained similar between 1983 [30] and 2000 [15] at 0.2 and 0.3 mg/100 g, respectively. No other stem or leafy vegetables were analysed at multiple timepoints.

#### 3.3.3. Fruiting Vegetables

In this vegetable group, eight vegetable types were analysed for iron content at two or more time points. Iron content remained stable with little change between 1983/1984 [31] and 2000 [15] for butternut pumpkin (0.4 and 0.2 mg/100 g), red (0.3 and 0.3 mg/100 g) and green capsicum (0.7 and 0.5 mg/100 g), and green zucchini (0.4 and 0.4 mg/100 g), and between 1983/1984 [31] and 2015 [41] for Lebanese cucumber (0.3 and 0.3 mg/100 g).

Mushroom, avocado and corn were additionally analysed at later timepoints. Button mushrooms ranged in iron content from 0.2 to 0.3 mg/100 g in 1983 [30] and 2000 [15], respectively, but higher in 2013 at 0.9 mg/100 g [37]. Hass avocado ranged from 0.5–0.7 mg/100 g between 1982 and 2015 [15,29,41], but higher values were reported in 2018 (2.0 mg/100 g) [49] and 2020 (1.0 mg/100 g) [50]. Corn showed the largest decline in iron content from 2.1 mg/100 g in 1983 [31] to 0.5 mg/100 g in 2000 [15] and 1.2 mg/100 g in 2017 [42].

#### 3.3.4. Brassica Vegetables

Iron content of broccoli, cauliflower and cabbage was reported at two or more time points. For broccoli, iron contents were relatively similar in 1982 [27] and 2000 [15] (1.0 and 0.8 mg/100 g, respectively). The iron content of cabbage was similar amongst varieties (red, savoy, white, green) and across time ranging from 0.4–0.6 mg/100 g from 1938 to 2000 [15,19,27]. Cauliflower iron concentrations reduced slightly from 0.6 mg/100 g in 1982 [27] to 0.4 and 0.3 mg/100 g in 2000 [15] and 2017 [42], respectively.

#### 3.3.5. Beans and Peas (Green Beans and Peas)

Green beans were first analysed in 1938 [19] with an iron content of 1.7 mg/100 g but lowered to 1.0 mg/100 g in both 1982 [25] and 2000 [15], and 0.7 mg/100 g in 2015 [41]. Similarly for green peas, iron content was highest in 1938 at 2.8 mg/100 g [19] and lower in 1982 [25] and 2000 [15] at 1.8 mg/100 g at both timepoints.

#### 3.3.6. Lentils (Dried)

Amongst Australian-grown lentils, red, green, and French varieties were identified and analysed between 1997 and 2016. The highest iron concentration was reported in 1997 at 65.5 mg/100 g (unspecified variety) [34] but ranged from 5.4 (French) [39], 5.6 (green) [39] and 9.1 (red) [39] and 5.5–9.0 mg/100 g (unspecified varieties) in the 2010s [46,47].

#### 3.3.7. Chickpeas (Dried)

Amongst chickpeas, both Desi and Kabuli varieties were analysed at multiple timepoints between 1997 and 2014. For Desi chickpea, iron levels ranged from 5.0 mg/100 g in 1997 [34] to 3.6–4.6 mg/100 g in 2010′s [39,45] and for Kabuli chickpea, iron levels tended to range from 5.7 mg/100 g in 1997 [34] to 3.8 mg/100 g in 2001 [38], and 4.1–4.9 mg/100 in 2010s [39,45].

#### 3.3.8. Other Legumes (Dried) (Peas, Fava bean, Mung Bean, Lupin)

Dried green peas were analysed between 1977 and 2016 on four occasions, with the highest iron content reported in 1977 (7.8 mg/100 g) [20], followed by 5.3 mg/100 g in 1997 [34], and 5.7–5.9 mg/100 g in 2010s [39,46]. Other varieties of dried peas (yellow, pigeon, cow pea) were only analysed at a single time point. Fava beans (broad beans) were analysed at multiple timepoints, and iron content ranged from 5.4–9.6 mg/100 g in 1997 [34,35] and 1998 [35] to 3.7 mg/100 g in 2012 [39]. In 2016, iron values of 10.5–13.0 mg/100 g were reported for fava beans but in highly manipulated experimental conditions [47]. For green mung beans, the iron content varied from 4.9 mg/100 g in 1997 [34] to 4.0 mg/100 g in 2012 [39], and 4.1–5.3 mg/100 in 2020 [48].

The most common lupins consumed by humans, sweet lupin (*Lupinus angustifolius)* and white lupin (*Lupinus albus*), were both analysed at multiple timepoints. The iron content of sweet lupin ranged from 6.0–19.6 mg/100 g in the 1970s [20,21,23], 5.2 mg/100 g in 1980s [32]; 4.6–6.9 mg/100 g in 1990s [33,34], to 2.7–5.4 mg/100 g in 2010s [40,43,46] and 2.8 mg/100 g in 2021 [51]. For white lupin, the iron content showed a decrease; from 5.0–11.4 mg/100 g in late 1970s [20,21,22,23], to 4.2–4.3 mg/100 g in 1980s [32], 2.7 mg/100 g in 1990s [34] and 2.5 mg/100 g in 2021 [51].

### 3.4. Changes in Iron content in Australian Food Composition Databases 1989–2019

Table 2 shows the changes in iron content as revealed in the Australian food composition databases for the vegetable groups and legumes for which data were available. For the majority of vegetables (35 out of 54, 65%) and all legumes (7 out of 7), iron content was only analysed at one timepoint and no temporal changes could be examined. For vegetables that were analysed at two or more time points (n = 19), the iron content decreased over time for six vegetables, remained the same (±0.1 to allow for rounding) for 12, and increased for one vegetable. The largest reduction in iron content was found for sweetcorn; from 2.1 to 1.2 mg/100 g from 1989 to 2019, followed by red-skinned potato (0.5 to 0.2 mg/100 g), cauliflower (0.6 to 0.3 mg/100 g) and green beans (1.0 to 0.7 mg/100). The only vegetable that revealed an increase in iron content was mushroom, from 0.2 to 0.5 mg/100 g between 1989 and 2019.

## 4. Discussion

This scoping review provides a comprehensive evaluation on the change in iron content of Australian vegetables and legumes using both published and grey literature data sources. Overall, the number of studies conducted were limited, with the first study published in 1938, followed by the majority of studies being conducted after 1980. Many vegetables and legumes were only analysed at one timepoint while others such as chickpeas and lupins were analysed at more than three different timepoints. Based on these data, our findings indicate that temporal changes in iron content of vegetables and legumes varied by individual type. For most vegetables, iron content remained relatively unchanged or decreased slightly. The vegetables with the largest declines over time included sweet corn, red-skinned potatoes, cauliflower, and fresh green beans (30–50%) while increases were noted for Hass avocadoes, mushrooms and silverbeet (150–300%). More pronounced reductions in iron content were observed for legumes, with early values, in particular pre-2000, typically being much higher and more variable than recent ones.

In addition, the changes in iron values based on vegetables and legumes included in six Australian food composition databases published between 1989 to 2019 were assessed. The majority were only analysed at one time point although a large number of vegetables were analysed during the early 1980 s for inclusion in the first Australian database and again in 2000 by FSANZ [15]. For vegetables analysed at more than one timepoint, most iron values were unchanged or slightly lower over time. The largest differences were seen in sweetcorn, red-skinned potato, and cauliflower, which approximately halved, and for mushroom which approximately doubled.

However, direct comparability of iron content among vegetable and legume sampled is not possible due to numerous differences in sampling such as geographic, environmental, and seasonal variability, variety of genotypes and cultivars sampled, and different analytical techniques used to measure iron content among studies. These factors have been shown to result in greater than 2-fold variability in iron content for vegetables and legumes [14].

Our studies sourced vegetables and legumes from different geographic regions of Australia, with exposure to variable climatic and soil conditions that are known to affect plant iron concentrations [52,53]. Differences in seasonal and growth conditions also affect iron content, for example, greenhouses and growth chambers offer more protection and control over climate, soil, crop health and pest infestations than field production [54]. This may explain the higher iron content values observed in our review for lupin grown in greenhouses in studies conducted before 1980 than in later field trials.

Use of different genotypes and cultivars during sampling and over time can contribute to variations in iron content of 2–4 fold [14]. For example, iron content varied 3–4 fold among samples and varieties of a single crop species such as soybeans (6–20 mg/100 g) and common beans (*Phaseolus vulgaris* 3.1–12.1 mg/100 g) [13,14,55,56]. In the vegetables and legumes reported here, specific cultivar names were often not reported and it is unknown whether or not commodities purchased at different times were the same cultivar or a new one, even if the same common name was used.

Analytical techniques have evolved over time with improvements in accuracy and precision. Early wet chemistry methods, such as colorimetry in the 1930s [19], were replaced with atomic absorption spectrophotometry (AAS) in the mid 1970s [14], and then ICP-Atomic Emission Spectrometry, ICP-Optical Emission Spectrometry and ICP-Mass Spectrometry after 1990. These analytical methods differ in their degree of extraction of iron from plant genotypes as well as their ability to identify contamination, with ICP methods having greater accuracy and specificity [14]. Since soils contain much higher concentrations of iron than plants [13], tiny amounts of unremoved soil can increase iron content values considerably. This may explain the higher iron values reported when older analytical methods were applied.

Considering the differences in sampling and analytical methods, it is likely that the variation in iron content of vegetables and legumes observed in this review (at most 2-fold), falls within expected natural ranges with no clear evidence of a temporal change. Previous studies examining changes in iron content of horticultural produce over time, have reported inconsistent results [12,13,57,58]. Mayer (1997) found a significant overall decrease of 22% between 1930s to 1980s in vegetables using the UK food composition database, although the presence of anomalies of measurement or sampling, changes in the food system, changes in the varieties grown or changes in agricultural practice could not be discounted [13]. Davis et al. (2004) found similar results (27% overall decrease) using US composition from 1950 to 1999 although approximately a quarter of garden crops tested showed an increase in iron content over time [14]. A re-analysis of these UK and US data by White and Broadley (2005), adjusting for moisture content found no statistical differences in iron content between 1930s and 1980s or later [55]. Similarly, Bruggraber et al. (2012) did not find significant changes in iron content when a group of vegetables in the UK were re-analysed in 2000 compared with food composition values from the 1930s and 1980s (difference of 11%) [56].

This scoping review has a number of strengths and limitations. A key strength of this study was the good coverage of data from both published and grey literature sources through extensive search on numerous electronic databases with no limitation on timeframe, relevant websites, and food composition databases. Several potentially eligible records were unavailable although extensive efforts were made to retrieve these through liaising with university library staff and contacting affiliated organisations. Direct comparability of iron content was not possible and no statistical analyses were conducted due to a paucity of available data, a lack of studies reporting some measure of iron content variability, and the large heterogeneity in sampling and analytical methods used.

It has been argued that comparison of historical food composition tables is not a reliable way to determine changes in nutrient composition of foods over time as the data may not be representative of domestically grown produce and is a reflection of the food supply available at the time [14]. Nevertheless, food composition data are used to estimate nutrient intakes of the population and it is therefore important to assess whether shifts have occurred over time in the nutrient content of staple foods consumed. As many vegetables and legumes are grown domestically in Australia, food composition data is thus more likely to reflect nutrient content of Australian produce. This highlights the need to maintain the currency of the database for monitoring the nutritional content of Australian produce. For example, the iron content of legumes and some commonly consumed leafy green vegetables have not been updated since the 1980s, and as plant-based diets are gaining popularity, this is likely to become increasingly relevant.

In the context of the Australian diet, vegetable and legume products/dishes on average contributed 9.6% and 1.1%, respectively, to the total iron intake of Australians aged two years and above [59]. Adults obtain more iron from vegetables and legumes than children (11.4% versus 7.8%), and females more than males (11.9% versus 9.7%) [59]. As both decreases and increases were noted in iron content of some commonly consumed vegetables, this is unlikely to affect the overall dietary iron intake at a population level. Although contribution of iron from vegetables and legumes could potentially be higher for vegetarians and vegans, any small decrease in iron content of vegetables and legumes could be easily managed through increasing consumption of widely available iron-fortified foods such as cereal and bread.

## 5. Conclusions

In conclusion, this scoping review provides a comprehensive evaluation on the change in iron content of Australian vegetables and legumes. It highlights a paucity of data on iron content over the past 100 years with most being collected between the 1980 and 2018. Based on the available limited data, and due to variations in sampling, analytical techniques and likely differences in growing location and season, no definitive temporal trends could be established. As more people are following plant-based or vegetarian diets, consistent monitoring of the nutrient composition of staple plant-based foods is strongly recommended.

## Figures and Tables

**Figure 1 foods-11-00056-f001:**
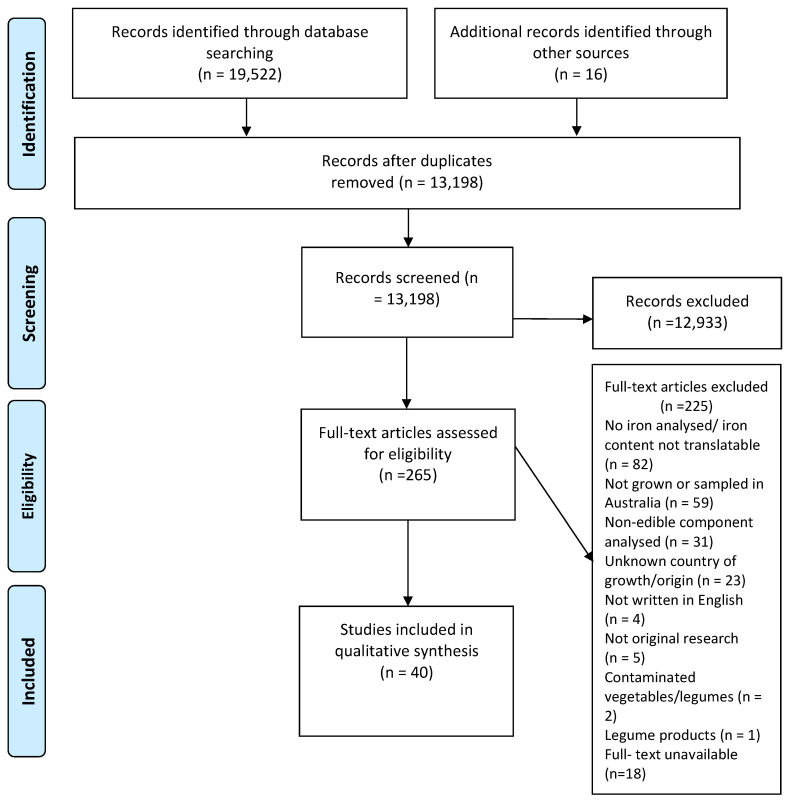
PRISMA flowchart illustrating the systematic screening process performed for the scoping review.

**Table 1 foods-11-00056-t001:** Key characteristics of studies reporting iron content of vegetables and legumes grown or sourced in Australia. Data were derived from raw, edible portion unless otherwise specified, rounded to the nearest one decimal place.

Reference	Food Type	Iron Content (mg/100 g) ^a^	Sampling Details (Number, Location and Collection Date)	Analytical Method
Lee et al., 1938 [19]	Bean, green	1.7 (AD 0.4)	4–12 samples obtained in QLD (Tewantin, Kingaroy, Innisfail, Malanda, Mt. Isa, Longreach, Charleville); date collected unknown; field trial	Thiocyanate colorimetry
	Cabbage	0.5 (AD 0.1)		
	Lettuce	1.0 (AD 0.4)		
	Pea, green	2.8 (AD 0.1)		
	Pumpkin	0.7 (AD 0.3)		
	Silverbeet	1.9 (AD 0.8)		
Hocking et al., 1977 [20]	Lupin, sweet, dried	19.6	Embryo component of 40–50 fruits across all three commodities; grown in Perth WA; greenhouse in sand culture; May-Nov, year unknown	AAS
	Lupin, white, dried	5.0		
	Pea, green dried	7.8		
Hocking et al., 1978 [21]	Lupin, sweet, dried	9.1	12 samples from 36 plants across 12 plots; grown in Perth WA; greenhouse in sand culture; Jul-Nov, year unknown	AAS
	Lupin, white, dried	5.6		
Pate et al., 1978 [22]	Lupin, white, dried	7.9 (95% CI 0.72)	3 samples of 10 fruits harvested 8 weeks after antithesis (control group); grown in Perth WA; greenhouse in sand culture; Aug-Nov, year unknown	AAS
Hocking 1980 [23]	Lupin, sweet, dried	6.0	3 samples of 100 seeds; grown in Perth WA, greenhouse in culture tanks, date collected unknown	AAS
	Lupin, white, dried	11.4		
Sale et al., 1980 [24]	Soya bean, dried	1.8	4 replicate samples of 10 seeds of similar age, grown in Camden NSW; field grown; Dec, 1979	AAS
Wills et al., 1984 [25]	Bean, green	1.0	Composite sample of produce purchased from 4 retail outlets in Sydney, grown in Australia ^b^; 1982	AAS
	Pea, green	1.8		
Wills et al., 1984 [26]	Potato, new	0.6	2 composite samples, purchased from retail outlets in Sydney, grown in Australia ^b^; 1982–1983	AAS
	Potato, Pontiac	0.5		
	Potato, Sebago	0.6		
Wills et al., 1984 [27]	Broccoli	1.0	Samples of each food item or 1 kg of smaller produce were obtained from 5 different retail outlets in Sydney; grown in Australia ^b^; 1982	AAS
	Brussels sprouts	0.9		
	Cabbage, red	0.6		
	Cabbage, savoy	0.6		
	Cabbage, white	0.6		
	Cauliflower	0.6		
	Kohlrabi	0.7		
Wills et al., 1984 [28]	Bean sprouts	0.4	A composite of 2 samples; purchased at market garden in Sydney; grown in Australia ^b^; 1982	AAS
	Bitter melon	0.9		
	Cabbage, Chinese	0.3		
	Cabbage, mustard	0.7		
	Choi sum	1.7		
	Hairy melon	0.3		
	Watercress	3.0		
Wills et al., 1986 [29]	Avocado, Hass	0.7	A composite of 5 samples; purchased from retail outlets in Sydney; grown in Australia ^b^; 1983	AAS
	Avocado, Fuerte	0.6		
Wills et al., 1986 [30]	Artichoke, globe	0.5	Samples of each food item or 1 kg of smaller produce were obtained from 5 different retail outlets in Sydney; grown in Australia ^b^; 1983	AAS
	Asparagus	1.0		
	Celery	0.2		
	Endive	1.7		
	Lettuce, common	0.6		
	Lettuce, cos	0.7		
	Lettuce, mignonette	1.1		
	Mushroom, button	0.2		
	Silverbeet	2.3		
	Spinach	3.2		
Wills et al., 1987 [31]	Bean, broad (fava)	1.9	Samples of each food item or 1 kg of smaller produce were obtained from 5 different retail outlets in Sydney; grown in Australia ^b^; 1983–1984 Samples of broad beans were fresh, not dried.	AAS
	Bean, butter (lima)	0.4		
	Bean, purple	1.2		
	Bean red (kidney)	2.4		
	Capsicum, green	0.7		
	Capsicum, red	0.3		
	Choko	0.3		
	Cucumber, green	0.1		
	Cucumber, Lebanese	0.3		
	Cucumber, telegraph	0.3		
	Eggplant	0.2		
	Pea, snow	0.9		
	Pumpkin, Butternut	0.4		
	Pumpkin, Golden	0.2		
	Nugget			
	Pumpkin, Qld Blue	0.9		
	Squash, Button	0.3		
	Squash, Scallopini	0.8		
	Sweet corn	2.1		
	Zucchini, Blackjack	0.6		
	Zucchini, Golden	0.4		
	Okra	1.1		
Hung et al., 1988 [32]	Lupin, sweet, dried	5.2 (R 2.6–7.6)	22 samples of sweet lupin seeds and 11 samples of white lupin seeds collected from different locations in Victoria; field grown, date collected unknown	ICP-AES
	Lupin, white, dried	4.3 (R 3.0–4.6)		
Bolland et al., 1993 [33]	Lupin, sweet, dried	4.6 (SE 0.3)	3 samples grown in South Carrabin WA; field grown, May 1987 3 samples grown in Badgingarra WA; field grown, May 1987	ICP-AES
	Lupin, sweet, dried	5.3 (SE 0.9)		
Petterson et al., 1997 [34]	Cowpea, dried Mung bean, green, dried Mung bean, black, dried Adzuki bean, dried Bean, fava, dried Pea, green dried Bean, navy, dried Bean, lima, dried Lupin, sweet, dried Lupin, white, dried Lentil, dried Chickpea, Kabuli, dried Chickpea, Desi, dried Pigeon pea, dried	5.9 (R 5.2–6.4) 4.9 (R 4.8–5.2) 6.6 (R 4.3–9.2) 4.4 9.6 (R 4.0–16.9) 5.3 (R 3.5–9.0) 6.8 (R 5.6–9.3) 5.7 6.9 (R 3.1–15.0) 2.7 (R 2.1–4.4) 65.5 (R 4.3–341) 5.7 (R 3.2–12.5) 5.0 (R 3.5–12) 3.3 (R 2.9–4.0)	1–535 samples collated from various sources, grown in Australia, date collected unknown	Various
Bolland et al., 2000 [35]	Bean, fava, dried	5.4 (SE 0.06) 5.7 (SE 0.07) 6.1 (SE 0.08)	Unspecified sample size, grown in South Kukerin WA 1997, East Pingaring WA 1998, and West Pingaring WA 1998, respectively; field grown	ICP-AES
Cunningham et al., 2002 [15]	Avocado, Hass	0.5	6–11 samples of each vegetable were purchased from different regions (VIC, SA, QLD, NSW, WA, TAS); 2000	ICP-MS
	Bean, green	1.0		
	Broccoli	0.8		
	Cabbage, green	0.4		
	Capsicum, green	0.5		
	Capsicum, red	0.3		
	Cauliflower	0.4		
	Celery	0.3		
	Cucumber, Greenridge	0.3		
	Lettuce, iceberg	0.6		
	Mushroom, button	0.3		
	Pea, green	1.8		
	Potato, Coliban	0.5		
	Potato, Desiree	0.2		
	Pumpkin, Butternut	0.2		
	Pumpkin, Jarrahdale	0.3		
	Sweet corn	0.5		
	Zucchini, green	0.4		
Cotching et al., 2011 [36]	Potato, Shepody	0.7	4 replicate samples, grown in Tasmania; 2007–2008; greenhouse	ICP-AES
	Silverbeet	3.0		
Koyyalamudi et al., 2013 [37]	Mushroom, button	0.9 (R 0.6–1.0)	6 samples; grown in Australia; field trial, date collected unknown,	ICP-MS
Wood et al., 2014 [38]	Chickpea, Desi, dried	5.4 (R 3.7–7.0)	Sample size not specified; grown in Breeza NSW; field grown, 2001	ICP-AES
	Chickpea, Kabuli, dried	3.8		
Broom et al., 2014 [39]	Chickpea, Desi, dried	4.6	1 sample for each food type, grown in GRDC classified Northern and Southern regions; field grown, 2012	ICP-AES/ICP-MS
	Chickpea, Kabuli, dried	4.1		
	Bean, fava, dried	3.7		
	Lentil, French, dried	5.4		
	Lentil, green, dried	5.6		
	Lentil, red, dried	9.1		
	Lupin, sweet, dried	4.0		
	Mung bean, green, dried	4.0		
	Pea, green, dried	5.9		
	Pea, yellow, dried	4.8		
FSANZ, 2015 [40]	Lupin, sweet, dried	3.9 4.3	1 sample, (with and without hull, respectively); grown in WA; field grown, 2014	ICP-AES/ICP-MS
FSANZ, 2015 [41]	Avocado, Hass	0.5	A composite of 8 samples of each vegetable; purchased from retail outlets in Australia, date purchased unknown	ICP-MS/ICP-AES
	Bean, green	0.7		
	Cucumber, Lebanese	0.3		
	Rocket	1.6 (R: 1.0–2.2)		
	Spinach, baby	1.8 (R: 1.0–2.7)		
FSANZ, 2017 [42]	Cauliflower	0.3	A composite of 8 samples; purchased from retail outlets in Australia, date purchased unknown	ICP-MS/ICP-AES
	Corn	1.2		
Karnpanit et al., 2017 [43]	Lupin, sweet, dried	2.7 (SD 0.70) 3.2 (SD 0.93)	10 samples of 10 different cultivars (with and without hull, respectively); grown in Wongan Hills Research Centre WA; field grown, 2011–2013	AAS
FSANZ, 2018 [44]	Kale	1.6	A composite of 8 samples; purchased from retail outlets in Australia, date purchased unknown	ICP-MS/ICP-AES
Tan et al., 2018 [45]	Chickpea, Desi, dried	4.2 (R 4.0–4.5)	3–5 samples each of 3 different cultivars; grown in Billa Billa Qld, Roma Qld, Warra Qld and Kingaroy Qld, respectively; field grown, Jun-Aug 2014 3–5 samples each of 3 different cultivars; grown in Billa Billa Qld, Roma Qld, and Warra Qld, respectively; field grown, Jun–Aug 2014	ICP-AES
		4.5 (R 4.1–4.9)		
		4.5 (R 4.2–4.7)		
		3.6 (R 3.3–4.2)		
	Chickpea, Kabuli, dried	4.4 (R 4.1–4.7)		
		4.9 (R 4.4–5.2)		
		4.6 (R 4.1–5.5)		
Zhang et al., 2018 [46]	Lentil, dried Lupin, sweet, dried Pea, green, dried	5.5 (SD 0.50) 5.4 (SD 0.55) 5.7 (SD 0.99)	3 samples for lentil and lupin; grown in Ouyen VIC; 2016; field trial 3 samples for pea; grown in Rokewood VIC; field trial, 2016	ICP-AES
Parvin et al., 2019 [47]	Bean, fava, dried	13.0 (SE 0.44)	4 samples for fava beans; grown in Horsham VIC in dry and wet conditions, respectively; experimental condition in growth chamber, May 2016 8 samples for lentils; grown in Horsham VIC; experimental in open top chamber; dry condition May 2015, and wet condition June 2016, respectively	ICP-AES
		10.5 (SE 0.18)		
	Lentil, dried	9.0 (SE 0.52)		
		6.2 (SE 0.28)		
Johnson et al., 2020 [48]	Mung beans, green, dried	5.3 (SD 0.1) 4.1 (SD 0.4)	2 brands (Pattu and Katoomba, respectively), 3 replicates from 2 digests each; grown in Australia; purchased from 2 Australian supermarkets, date purchased unknown	ICP-MS
Perkins et al., 2020 [49]	Avocado, Hass	2.0	A composite sample; grown in Adare, Queensland; commercial orchard, 2018	ICP-AES
Kämper et al., 2021 [50]	Avocado, Hass	1.0 (SD 0.5)	95 samples, Childers, Queensland; commercial orchard, April-May 2018	ICP-AES
Mazumder et al., 2021 [51]	Lupin, sweet, dried	2.8 (SD 3.4)	Mean of 6 cultivars, 3 replicates each, grown in Wagga Wagga, New South Wales; year collected unknown Mean of 3 cultivars, 3 replicates each, grown in Wagga Wagga, New South Wales; year collected unknown	ICP-OES
	Lupin, white, dried	2.5 (SD 4.2)		

^a^ Values expressed as mean with or without a measure of variability depending on the reference. ^b^ Purchased in Sydney with growth location not specified, however samples were considered as Australian grown given that >98% of total vegetables available at the time were locally produced. Abbreviations: AAS: atomic absorption spectroscopy; AD: average deviation; CI: Confidence Interval; FSANZ: Food Standards Australia New Zealand; ICP-AES: inductively coupled plasma atomic emission spectroscopy; ICP-MS: Inductively coupled plasma-mass spectrometry; ICP-OES: Inductively coupled plasma optical emisssion spectroscopy; R: range; SD: standard deviation; SE: standard error.

**Table 2 foods-11-00056-t002:** Iron content of vegetables and legumes reported in Australian reference food composition databases between 1989 to 2019.

Figure A1989.	Processing	CoFA1989	NUTTAB1991–2	NUTTAB1995	NUTTAB2006	NUTTAB2010	AFCD2019
**Potatoes**							
Potato, new	Raw	0.6	0.6	0.6	0.6	0.6	0.6
Potato, Coliban (pale skin)	Raw	na	na	na	0.4	0.5	0.5
Potato, Sebago (pale skin)	Raw	0.6	na	na	0.6	0.6	0.6
Potato, pale skin (unspec)	Raw	0.6	0.6	0.6	0.6	0.5 ^a^	0.5
Potato, Desiree (red skin)	Raw	na	na	na	0.2	0.2	0.2
Potato, Pontiac (red skin)	Raw	0.5	na	na	na	0.5	0.5
Potato, red skin (unspec)	Raw	0.5	0.5	0.5	0.3	0.2 ^a^	0.2
**Stem and leafy vegetables**							
Artichoke, globe	Raw	0.5	0.5	0.5	0.5	0.5	0.5
Asparagus	Raw	1.0	1.0	1.0	1.0	1.0	1.0
Celery	Raw	0.2	0.2	0.2	0.2	0.3 ^a^	0.3
Endive	Raw	1.7	1.7	1.7	1.7	1.7	1.7
Lettuce, cos	Raw	0.7	0.7	0.7	0.7	0.7	0.7
Lettuce, iceberg	Raw	0.6	0.6	0.6	0.6 ^a^	0.6	0.6
Lettuce, mignonette	Raw	1.1	1.1	1.1	1.1	1.1	1.1
Silverbeet	Raw	2.3	2.3	2.3	2.5	2.3	2.3
Spinach, English	Raw	3.2	3.2	3.2	3.5	3.2	3.2
Spinach, water	Raw	2.4	2.4	2.4	2.4	2.4	2.4
Watercress	Raw	3.0	3.0	3.0	3.0	3.0	3.0
**Fruiting vegetables**							
Avocado	Raw	0.7	0.7	0.7	0.6 ^a^	0.5	0.5 ^a^
Capsicum, green	Raw	0.7	0.7	0.7	0.6	0.6 ^a^	0.5 ^a^
Capsicum, red	Raw	0.3	0.3	0.3	0.3	0.3 ^a^	0.3
Choko	Raw	0.3	0.3	0.3	0.3	0.3	0.3
Cucumber, common	Raw	0.1	0.1	0.1	0.2 ^a^	0.2	0.2
Cucumber, Lebanese	Raw	0.3	0.3	0.3	0.3	0.3	0.3 ^a^
Cucumber, telegraph	Raw	0.3	0.3	0.3	0.3	0.3	0.3
Eggplant	Raw	0.2	0.2	0.2	0.2	0.2	0.2
Melon, bitter	Raw	0.9	0.9	0.9	0.9	0.9	0.9
Melon, hairy	Raw	0.3	0.3	0.3	0.3	0.3	0.3
Mushroom	Raw	0.2	0.2	0.2	0.3 ^a^	0.3	0.45 ^a^
Okra	Raw	1.1	1.1	1.1	na	1.1	1.1
Pumpkin, Butternut	Raw	0.4	0.4	0.4	0.3 ^a^	0.3	0.3
Pumpkin, Golden Nugget	Raw	0.2	0.2	0.2	0.2	0.2	0.2
Pumpkin, Jarrahdale	Raw	na	na	na	0.1	0.1	0.1
Pumpkin, Queensland Blue	Raw	0.9	0.9	0.9	0.9	0.9	0.9
Pumpkin, unspec	Raw	0.5	0.5	0.5	0.2	0.4	0.4
Squash, button, yellow	Raw	0.3	0.3	0.3	0.3	0.3	0.3
Squash, scallopini, green	Raw	0.8	0.8	0.8	0.8	0.8	0.8
Sweet corn	Raw	2.1	2.1	2.1	1.0 ^a^	1.0	1.2 ^a^
Zucchini, green	Raw	0.6	0.6	0.6	0.5 ^a^	0.5	0.5
Zucchini, golden	Raw	0.4	0.4	0.4	0.4	0.4	0.4
**Brassica vegetables**							
Broccoli	Raw	1.0	1.0	1.0	0.9 ^a^	0.8	0.8
Brussels sprout	Raw	0.9	0.9	0.9	0.9	0.9	0.9
Cabbage, Chinese	Raw	0.3	0.3	0.3	0.3	0.3	0.3
Cabbage, Chinese flowering	Raw	1.7	1.7	1.7	1.7	1.7	1.7
Cabbage, mustard	Raw	0.7	0.7	0.7	0.7	0.7	0.7
Cabbage, red	Raw	0.6	0.6	0.6	0.6	0.6	0.6
Cabbage, savoy	Raw	0.6	0.6	0.6	0.6	0.6	0.6
Cabbage, white/common	Raw	0.6	0.6	0.6	0.5 ^a^	0.5	0.5
Cauliflower	Raw	0.6	0.6	0.6	0.5 ^a^	0.5	0.3 ^a^
Kohlrabi	Raw	0.7	0.7	0.7	0.7	0.7	0.7
**Beans and peas (green)**							
Bean, green	Raw	1.0	1.0	1.0	1.1 ^a^	1.1	0.7 ^a^
Bean sprouts	Raw	0.4	0.4	0.4	0.4	0.4	0.4
Pea, green	Raw	1.8	1.8	1.8	1.8 ^a^	1.8	1.8
Pea, snow	Raw	0.9	0.9	0.9	0.9	0.9	0.9
**Legumes**							
Bean, fava	Raw (fresh)	1.9	1.9	1.9	1.9	1.9	1.9
Bean, haricot	Dried	na	6.4	6.4	6.4	6.4	6.4
Bean, kidney	Dried	na	5.6	5.6	5.6	5.6	5.6
Bean, lima	Dried	na	5.7	5.7	5.7	5.7	5.7
Lentil, red, brown, green	Dried	na	7.5	7.5	7.5	6.7	6.7
Pea, split	Dried	na	3.8	3.8	3.8	3.8	3.8
Soya bean	Dried	na	9.5	9.5	9.5	9.5	na

^a^ the time when iron value were obtained from a new sample analysis. AFCD2019: Australian Food Composition Database Release 1 (2019); CoFA1989: Composition of Foods Australia (1989); na: not available; unspec: unspecified variety, estimate of average values.

## Data Availability

Data is contained within the article or Appendix A.

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
