# Peer review of "Temporal Change in Iron Content of Vegetables and Legumes in Australia: A Scoping Review"

_foods, 2021, doi:10.3390/foods11010056_

Round 1

Reviewer 1 Report

In this study, the authors aimed to provide a comprehensive evaluation of the temporal change in the iron content of Australian vegetables and legumes. I have some minor and major comments.

  1. Please remove Reference 1 and add a more recent and more relevant reference.
  2. Please add the name of the first author to the list for each paper in Table 1.
  3. Please define the abbreviations in footnotes for Table 2.
  4. How did the authors determine if a difference between iron content of vegetables and legumes over time was notable? For example, the authors stated that the Iron content of lettuce remained relatively stable over three time points but later we observe that the iron content has decreased from 1.0 mg/100g in 1938 to 0.6 mg/100g in both 1983 and 2000!
  5. The main concern that I have for this scoping review is that the iron content of vegetables and legumes depends heavily on environmental factors (e.g., type of soil, weather conditions, season, agronomic factors, storage conditions), genetic factors, and type of analytical method used for determination. How did the authors take these variations into account while comparing the iron content over time? The iron content of the same plant species may differ between two different locations at the same time due to differences in environmental factors and change in time cannot solely explain the variation in iron content of specific plant species.
  6. In the discussion section, the authors explained that differences in environmental conditions, variety of genotypes, and different analytical techniques used to measure iron content result in variability in iron content. Later, they also mentioned that due to the differences in sampling and analytical methods, variation in iron content of vegetables and legumes may fall within expected natural ranges with no clear evidence of a temporal change. Thus, it is important to discuss what factors are responsible for low iron content in specific plant species and how we can increase the iron content by modifying the environmental factors. In other words, what factors have changed over time which resulted in variation in iron content?

Author Response

Reviewer 1

In this study, the authors aimed to provide a comprehensive evaluation of the temporal change in the iron content of Australian vegetables and legumes. I have some minor and major comments.

  1. Please remove Reference 1 and add a more recent and more relevant reference.

This reference has been updated: ‘Bowman, S.A. A Vegetarian-Style Dietary Pattern Is Associated with Lower Energy, Saturated Fat, and Sodium Intakes; and Higher Whole Grains, Legumes, Nuts, and Soy Intakes by Adults: National Health and Nutrition Examination Surveys 2013-2016. Nutrients 2020, 12, doi:10.3390/nu12092668.’

  1. Please add the name of the first author to the list for each paper in Table 1.

Thank you, this has been amended.

  1. Please define the abbreviations in footnotes for Table 2.

Abbreviations have now been added to Table 2 footnotes.

  1. How did the authors determine if a difference between iron content of vegetables and legumes over time was notable? For example, the authors stated that the Iron content of lettuce remained relatively stable over three time points but later we observe that the iron content has decreased from 1.0 mg/100g in 1938 to 0.6 mg/100g in both 1983 and 2000!

Thank you for picking this up, we have now amended the text (Lines 203-205): ‘Iron content of lettuce was reported over three time points and ranged from 1.0 mg/100g in 1938 (unspecified variety) [19], to 0.6 mg/100g in both 1983 [30] and 2000 (common/iceberg) [15].’

The iron content of vegetables and legume types over time were mostly summarised as the mean values reported in the studies, with differences noted if they appeared to increase or decrease over time but no statistical analyses were undertaken due to limited studies and high variability between studies.

  1. The main concern that I have for this scoping review is that the iron content of vegetables and legumes depends heavily on environmental factors (e.g., type of soil, weather conditions, season, agronomic factors, storage conditions), genetic factors, and type of analytical method used for determination. How did the authors take these variations into account while comparing the iron content over time? The iron content of the same plant species may differ between two different locations at the same time due to differences in environmental factors and change in time cannot solely explain the variation in iron content of specific plant species.

We agree that it is very difficult to determine whether there were genuine changes in iron content over time, due to the many differences between studies, such as sampling, cultivars tested, analytical methods and environmental factors. Consequently, we have been very careful in the interpretation our findings. We have emphasised all the challenges with interpreting these data in detail in the Discussion, and also mention these in the Abstract and Conclusion:

Lines 312-342: ‘However, direct comparability of iron content among vegetable and legume sampled is not possible due to numerous differences in sampling such as geographic, environmental, and seasonal variability, variety of genotypes and cultivars sampled, and different analytical techniques used to measure iron content among studies. These factors have been shown to result in greater than 2-fold variability in iron content for vegetables and legumes [14].’

‘Our studies sourced vegetables and legumes from different geographic regions of Australia, with exposure to variable climatic and soil conditions that are known to affect plant iron concentrations [53,54]. Differences in seasonal and growth conditions also affect iron content, for example, greenhouses and growth chambers offer more protection and control over climate, soil, crop health and pest infestations than field production [55]. This may explain the higher iron content values observed in our review for lupin grown in greenhouses in studies conducted before 1980 than in later field trials.’

‘Use of different genotypes and cultivars during sampling and over time can contribute to variations in iron content of 2-4 fold [14]. For example, iron content varied 3-4 fold among samples and varieties of a single crop species such as soybeans (6-20 mg/100g) and common beans (Phaseolus vulgaris 3.1-12.1 mg/100g) [13,14,56,57]. In the vegetables and legumes reported here, specific cultivar names were often not reported and it is unknown whether or not commodities purchased at different times were the same cultivar or a new one, even if the same common name was used.’

‘Analytical techniques have evolved over time with improvements in accuracy and precision. Early wet chemistry methods, such as colorimetry in the 1930s [19], were replaced with atomic absorption spectrophotometry (AAS) in the mid 1970s [14], and then ICP-Atomic Emission Spectrometry, ICP-Optical Emission Spectrometry and ICP-Mass Spectrometry  after 1990. These analytical methods differ in their degree of extrac-tion of iron from plant genotypes as well as their ability to identify contamination, with ICP methods having greater accuracy and specificity [14]. Since soils contain much higher concentrations of iron than plants [13], tiny amounts of unremoved soil can in-crease iron content values considerably. This may explain the higher iron values re-ported when older analytical methods were applied.’

  1. In the discussion section, the authors explained that differences in environmental conditions, variety of genotypes, and different analytical techniques used to measure iron content result in variability in iron content. Later, they also mentioned that due to the differences in sampling and analytical methods, variation in iron content of vegetables and legumes may fall within expected natural ranges with no clear evidence of a temporal change. Thus, it is important to discuss what factors are responsible for low iron content in specific plant species and how we can increase the iron content by modifying the environmental factors. In other words, what factors have changed over time which resulted in variation in iron content?

From our scoping review, we have determined that the studies examining iron content of vegetables and legumes in Australia over the past 80 years are highly heterogeneous, and we have outlined these factors above. Over time, analytical techniques have improved, new cultivars and genotypes have been developed and environmental conditions may have changed in various parts of Australia, thus making it very difficult to pinpoint exactly how these factors influence changes in iron content. To do so would require carefully conducted time series of studies.

Reviewer 2 Report

The manuscript aimed to perform a scoping review to examine the temporal changes of iron content in vegetables and legumes drawing on scientific and grey literature, and explore any iron content changes for these foods in the national food composition databases over time. The theme is interesting, but it is necessary to perform the database search again since it was performed in September 2020. Also, my major concern is related to the difficulty to compare data from vegetables in which is known that there are differences regarding “geographic, environmental, and seasonal variability, variety of genotypes and cultivars sampled, and different analytical techniques used to measure iron content among studies”, as mentioned in the discussion section. In addition, I have some comments indicated below.

Abstract: lines 22 – 24: vegetarianism is not in the context of the abstract since you did not mention it in the abstract background. Also, all the population can suffer from vegetables iron reduction, not only vegetarians.

Please, insert the study hypothesis/study question in the introduction section.

Considering the systematic approach, the search must be conducted again to include data from September 2020 until now.

Did you analyze the risk of bias?

Figure 1 is confusing. Adjust the columns and lines to separate each topic.

Lines 170-178 – mention that the data is in table 1.

Lines 180-181 - mention that the data is in table 1.

Lines 184-253 – Insert the references.

Table 1 -  If you mention the author name and year of publication in the first column it will be easier to read. The studies should be organized chronologically. It is lacking the years in some references.

Table 2 – insert all acronyms as a footnote

Lines 303-308 – It should be highlighted as a study limitation, as well as analytical techniques.

Thank you for the opportunity to review this manuscript!

Reviewer 3 Report

Please see below for suggestions aimed at improving the clarity and context of this manuscript.

  1. Table 1 - define AD, SD and SE for clarity in Table caption.
  2. Discussion - the authors have flagged the relative change in iron content from different studies and time-points. However, these have not been contextualised in relation to nutrient-based recommendations in Australia. Large percentage changes in infrequently-consumed food items eaten in small portions may be inconsequential in meeting these recommendations. Similarly, small changes in frequently consumed items may be much more important. I suggest contextualising your findings based on both national (sex, age and population-specific) guidelines as well as common dietary practices.

Round 2

Reviewer 1 Report

Thanks to the authors for revising the manuscript. I have no more comments.

Reviewer 2 Report

I congratulate the authors for the well-improvement of the manuscript!